# Whole genome sequence analysis showing unique SARS-CoV-2 lineages of B.1.524 and AU.2 in Malaysia

**Ummu Afeera Zainulabid**[1,2◉], **Aini Syahida Mat Yassim**[3◉], **Mushtaq Hussain**[4]*,
**Ayesha Aslam**[4], **Sharmeen Nellisa Soffian**[1], **Mohamad Shafiq Mohd Ibrahim**[5],
**Norhidayah Kamarudin**[6], **Mohd Nazli Kamarulzaman**[7], **How Soon Hin**[2]*, **Hajar
Fauzan Ahmad**[1,8]*

**1** Faculty of Industrial Sciences and Technology, Universiti Malaysia Pahang, Gambang, Pahang, Malaysia,
**2** Department of Internal Medicine, Kulliyyah of Medicine, International Islamic University of Malaysia,
Kuantan, Pahang, Malaysia, **3** Biovalence Sdn. Bhd., Petaling Jaya, Selangor, **4** Bioinformatics and
Molecular Medicine Research Group, Dow Research Institute of Biotechnology and Biomedical Sciences,
Dow College of Biotechnology, Dow University of Health Sciences, Karachi, Pakistan, **5** Department of
Paediatric and Dental Public Health, Kulliyyah of Dentistry, International Islamic University Malaysia,
Kuantan, Pahang, Malaysia, **6** Department of Pathology and Laboratory Medicine, Kulliyyah of Medicine,
International Islamic University of Malaysia, Kuantan, Pahang, Malaysia, **7** Department of Surgery, Kulliyyah
of Medicine, International Islamic University of Malaysia, Kuantan, Pahang, Malaysia, **8** Centre for Research
in Advanced Tropical Bioscience (Biotropic Centre), Universiti Malaysia Pahang, Gambang, Pahang,
Malaysia

◉ These authors contributed equally to this work.
* fauzanahmad@ump.edu.my (HFA); joanchua@iium.edu.my (HSH); mushtaq.hussain@duhs.edu.pk (MH)

doi.org/10.1371/journal.pone.0263678

SAUDI ARABIA

**Data Availability Statement:** All sequences data
are available at https://trace.ncbi.nlm.nih.gov/
Traces/sra/?study=SRP324679.

## Abstract

SARS-CoV-2 has spread throughout the world since its discovery in China, and Malaysia is
no exception. WGS has been a crucial approach in studying the evolution and genetic diver-
sity of SARS-CoV-2 in the ongoing pandemic. Despite considerable number of SARS-CoV-
2 genome sequences have been submitted to GISAID and NCBI databases, there is still
scarcity of data from Malaysia. This study aims to report new Malaysian lineages of the
virus, responsible for the sustained spikes in COVID-19 cases during the third wave of the
pandemic. Patients with nasopharyngeal and/or oropharyngeal swabs confirmed COVID-19
positive by real-time RT-PCR with $C_T$ value < 25 were chosen for WGS. The selected
SARS-CoV-2 isolates were then sequenced, characterized and analyzed along with 986
sequences of the dominant lineages of D614G variants currently circulating throughout
Malaysia. The prevalence of clade GH and G formed strong ground for the presence of two
Malaysian lineages of AU.2 and B.1.524 that has caused sustained spikes of cases in the
country. Statistical analysis on the association of gender and age group with Malaysian line-
ages revealed a significant association ($p < 0.05$). Phylogenetic analysis revealed dispersion
of 41 lineages, of these, 22 lineages are still active. Mutational analysis showed presence of
unique G1223C missense mutation in transmembrane domain of the spike protein. For bet-
ter understanding of the SARS-CoV-2 evolution in Malaysia especially with reference to the
reported lineages, large scale studies based on WGS are warranted.

**Funding:** The authors received funding for this work from the Ministry of Higher Education Malaysia and Sultan Ahmad Shah Medical Centre @ IIUM for supporting this work. Hajar Fauzan Ahmad awarded for FRGS/1/2019/WAB13/UMP/03/1 and Ummu Afeera Zainulabid is Principal Investigator for SRG21-040-0040, respectively. The funders had no role in study design, data collection and analysis, decision to publish, or preparation of the manuscript.

**Competing interests:** The authors have declared that no competing interests exist.

## Introduction

The emergence of Severe Acute Respiratory Syndrome Coronavirus 2 (SARS-CoV-2) in Wuhan, China in December 2019 resulted in an unprecedented global outbreak and soon recognized as pandemic, referred to as COVID-19 [1–4]. SARS-CoV-2 actively propagates in lungs as a primary site of infection. This active propagation leads to the storm of inflammatory cytokines that if not curtailed advances the pathology of the disease [5]. By November 2021, more than 260 million confirmed cases of COVID-19, with over 5 million deaths have been reported by the World Health Organization (WHO) [6]. By the same date, the cumulative number of confirmed cases of COVID-19 in Malaysia has reached over 2.6 million, of which over 30,000 died from the disease. The daily number of confirmed cases of COVID-19 has continued to soar with more than 10,000 cases per day since July, 2021, however at present, due to the mass vaccination drive, the cases are dwindling with encouraging pace [7]. Malaysia is facing a much tougher task in curbing the COVID-19 pandemic in its third wave which began on September 8, 2020 due to the Benteng LD cluster in Sabah [8]. Since then, the highest lineage contributor during the third wave of pandemic appeared to be B.1.524, with D614G and A701V mutations in the spike protein of the virus [9].

The WHO defined SARS-CoV-2 Variant of Concerns (VOCs) as variants with clear evidence indicating significant impact on transmissibility, severity (including hospitalizations or death) and/or immunity due to significant reduction in neutralization by antibodies generated during previous infection or by vaccination. This in total may impact the epidemiological landscape of the virus [10,11]. Whereas Variant of Interests (VOIs) are variants with specific genetic markers that have been associated with changes in receptor binding regions of the virus, reduced neutralization by antibodies generated due to previous infection or by vaccination, reduced efficacy of treatments, potential diagnostic impact, or predicted increase in transmissibility or disease severity, but warrant continuous monitoring and further investigations [12]. The SARS-CoV-2 VOC, α (B.1.1.7) was first detected in Malaysia in February 2021 followed by then considered VOI, η (B.1.525) VOC, β (B.1.351) in March this year. Whereas, SARS-CoV-2 VOC, Δ (B.1.617.2) and then considered VOI, κ (B.1.617.1), were first detected in June 2021 in Peninsular Malaysia [13]. In Sarawak, the first Δ variant was detected on June 2021, along with then recognized VOI, θ (P.3) in 2021 [14].

Of interest, all of the VOCs and VOIs detected in Malaysia harbour D614G mutation in their spike protein [15,16]. Until recently, 90.30% of all COVID-19 infection in Malaysia has been due to the D614G variant, and this mutation is present in all the new emerging variants [15]. As a result of positive natural selection, it was found that D614G increases the infectivity, viral fitness, transmission rate and efficiency of cellular entry for the SARS-CoV-2 virus across a broad range of human cell types [9,17–23]. Nevertheless, D614G mutation alone has not been shown to cause higher COVID-19 mortality or clinical severity, or alter the efficiency of the current laboratory diagnostic, therapeutics, vaccines or public health prevention strategies [11,24]. Therefore, in this study, we analyzed the dominant lineages of D614G variants currently circulating in Malaysia using whole genome sequences of the Malaysian SARS-CoV-2 deposited to the Global Initiative on Sharing All Influenza Data (GISAID) database. This study reports new Malaysian lineages that are responsible in causing sustained spikes in COVID-19 cases throughout the third wave of the pandemic in Malaysia. We have also investigated the divergence of the D614G variant of the Pahang SARS-CoV-2 isolates and explored its possible origin. Finally, we have computationally predicted possible effects of the G1223C mutation, observed in this study, resided in the transmembrane domain of the spike protein and uniquely detected in SARS-CoV-2 from Pahang, Malaysia.

## Materials and methods

### Sample selection

Nasopharyngeal and oropharyngeal swab test results from over 1000 patients that were confirmed positive for SARS-CoV-2 through real-time reverse transcriptase-PCR (real-time RT-PCR) at the Sultan Ahmad Shah Medical Centre were initially taken into consideration. Out of these only 10 patients were selected for whole genome sequencing of the virus on the basis of $C_T$ value <25 and when the total extracted genomic RNA level was found to be more than 10 ng/µl. Ethical approval (IREC 2021–080) for the study was obtained from IIUM Research Ethics Committee.

### RNAs extraction

Total genomic RNA was extracted using Maxwell HT simplyRNA kit (Promega, USA) following manufacturer guidelines.

### Next-generation sequencing of the full-length viral genome

Next-generation sequencing (NGS) library was constructed after amplifying full length genome using synthesized cDNA from SuperScriptIV (ThermoFisher Scientific, USA) with some modifications [25,26]. Briefly, 5 µl of the cDNA was used as template for multiplex PCR using Q5 polymerase (NEB, USA) as well as the Artic v3 primer pools during library preparation. The constructed library was then sequenced on an iSeq 100 System (Illumina, USA) (with run configuration of $1 \times 300$ bp).

### Sequence analysis

The SARS-CoV-2 genome was reconstructed from the raw reads using a combination of several bioinformatic tools enlisted in https://github.com/CDCgov/SARS-CoV-2_Sequencing/tree/master/protocols/BFX-UT_ARTIC_Illumina. Genome sequences from other studies related to humans and animal coronaviruses were mined from the GISAID (https://www.gisaid.org) and NCBI GenBank (https://www.ncbi.nlm.nih.gov/genbank/).

### Public database SARS-CoV-2 genome analysis

To study the dominant lineages and D614G frequency, a total of 1356 complete genome sequences of SARS-CoV-2 of Malaysian origin, submitted to GISAID from March 1, 2020 to July 19, 2021 were retrieved (S1 Table). Sequences were selected based on completion of the genome with minimum number of unresolved nucleotides. Restraining the selection from 1356 to 986 sequences (S1 File). Analysis of lineage distribution and clade frequency were performed manually by using Pivot table in Microsoft Excel. Real-time Malaysia SARS-CoV-2 Genomics Surveillance updates were monitored *via* (https://bit.ly/2UEFFGt).

The first virus from each lineage with D614G mutation in spike protein was extracted using patient's status metadata downloaded from GISAID on July 19, 2021. To do this, 1356 viruses were analysed manually using Pivot table and the dates were filtered to months and year in Microsoft Excel. The lineage description was classified according to the PANGO Lineage List (https://cov-lineages.org/lineage_list.html).

### Phylogenetic analysis

A total of selected 986 complete whole genome sequences of Malaysian variants with D614G mutation were retrieved from GISAID database (S1 Table; S1 File). A complete genome of

Wuhan-Hu-1 (NC_045512) was downloaded from GenBank (https://www.ncbi.nlm.nih.gov/sars-cov-2/) for outgroup. The multiple sequence alignment was performed using Clustal Omega [27] and observed in BioEdit [28] and finalized using MEGA XI [29].

Evolutionary analysis was conducted in MEGA XI by reconstructing bootstrap consensus tree of sequences employing Neighbor-Joining (NJ) method with 1000 bootstrap replicates to represent the evolutionary history of the taxa analyzed [30]. Branches corresponding partitions that are reproduced in less than 50% bootstrap replicates were collapsed. The evolutionary distances were computed using the Kimura 2-parameter method and are in the units of the number of base substitutions per site. The rate variation among sites was modelled employing gamma distribution (shape parameter = 1). All ambiguous positions were removed for each sequence pair employing complete deletion option).

## Mutation analysis

Mutation analyses were carried out using Nextclade v.1.5.2, a web-based analysis server (https://clades.nextstrain.org) by comparing against a wild-type of Wuhan-Hu-1 (NC_045512.2).

To evaluate the effect of mutations, a 3D structure model of wild-type spike protein (YP_009724390.1) was first generated using SWISS-model based on the most fitted protein template PDB ID: 6XR8 covering 14–1162 amino acids of the protein. For analyzing the effect of amino acid substitution in the TM domain, a 3D structure, PDB ID: 7LC8 (SARS-CoV-2 Spike protein TM domain) was used. Both 3D structure model of YP_009724390.1 and 7LC8 were uploaded to mCSM-PPI2 server [31]. Next, the potential pathogenic effect of the amino acid substitution on TM domain was investigated by uploading a 3D structure of TM domain, PDB ID: 7LC8 and TM domain amino acid sequence onto mCSM-membrane [32]. Similarly, Protein Variation Effect Analyzer (PROVEAN) [33] and SNAP 2 tools [34]; the web-based servers for predicting the effect of mutations were also used for the same purpose. The servers predicted the consequence of amino acid mutation to be whether benign or pathogenic, deleterious or neutral, effect or neutral, respectively.

## Statistical analysis

Data were presented as count and percentage. Chi-square test was carried out using IBM SPSS v25.0 to test the statistical significance for association of gender, patient status and age groups with Malaysian lineages. All level of significances were set at $p < 0.05$.

## Results

### Evolution of D614G variant of SARS-CoV-2 in the Malaysian population

Of the 1,502 SARS-CoV-2 complete genomes deposited to GISAID database, 1,356 contained spike D614G mutation in their genomes. To better characterize the local distribution of lineages that may contribute to the constant increase in COVID-19 cases in Malaysia, Fig 1 summarizes the distribution of the D614G variant lineages throughout the country since it was first detected on March 21, 2020. Based on the GISAID database analysis, there were 41 lineages of D614G variant dispersed throughout Malaysia. Lineage B.1.524 (n = 419) and AU.2 (n = 311) appeared to have caused significant transmission of the virus locally compared to Variant of Concern (VOC) α B.1.1.7 (n = 11), β (B.1.351(n = 161), B.1.351.3 (n = 2)) and Δ B.1.617.2 (n = 58); Variant of Interest (VOI) η B.1.525 (n = 3), κ B.1.617.1 (n = 3), as well as lineages currently designated alerts for further monitoring, such as P.2 (n = 1), P.3 (n = 10), B.1.466.2 (n = 70), B.1.214.2 (n = 1).

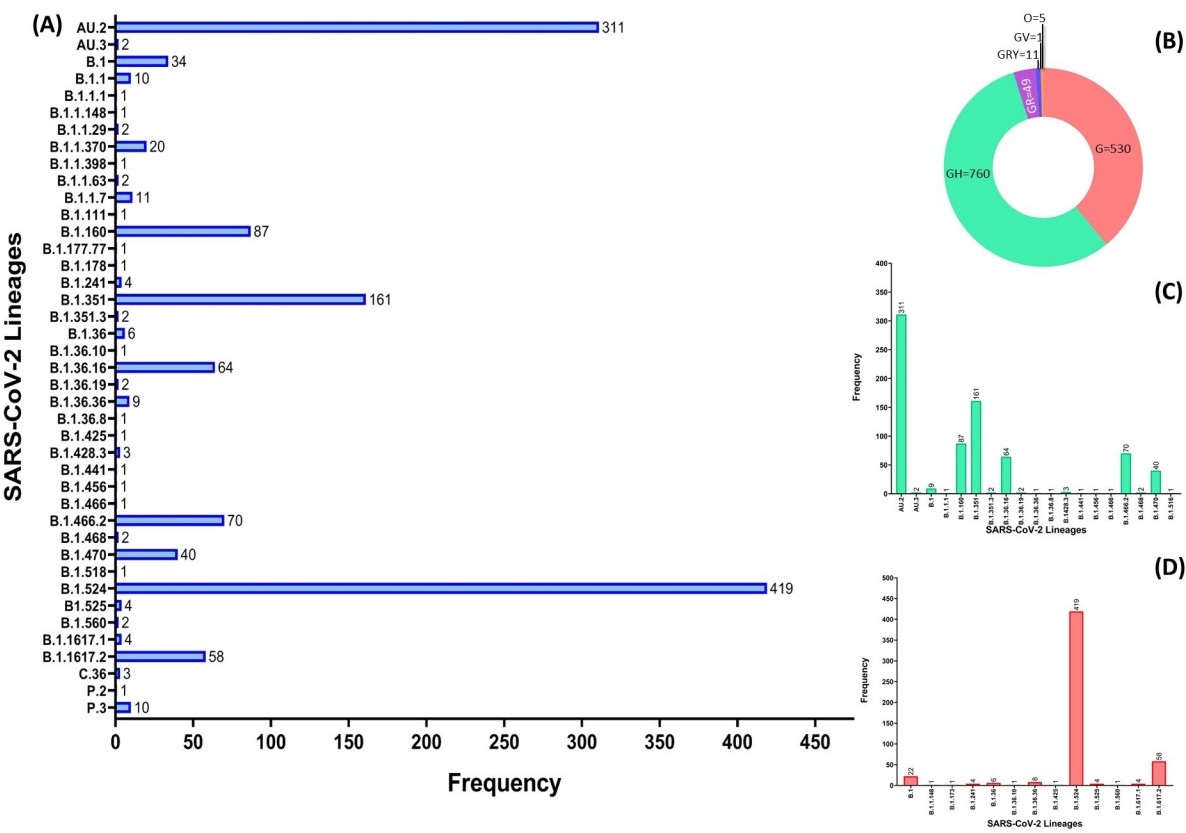

**Fig 1. SARS-CoV-2 D614G variant lineages and clades distribution in Malaysia.** (A) Distribution of lineages of SARS-CoV-2 D614G variant from Malaysia deposited in GISAID until July 5, 2021 (n = 1356). (B) Donut diagram showing clade distribution of D614G variants from Malaysia deposited in GISAID until July 19, 2021 (C) Lineages clustered in clade GH (D) Lineages clustered in clade G.

Next, we investigated the frequency of D614G variant clades circulating in Malaysia until July 2021. It appeared that of the six D614G variant clades (GH, G, GR, GRY, O and GV), GH makes up the largest clade with 760 of genomes from different lineages (Fig 1B). Further analysis of clade GH shows lineage AU.2 had appeared most often in the transmission of the disease (Fig 1C) followed by clade G, in which lineage B.1.524 seems to be the highest contributor in the local transmission of COVID-19 (Fig 1D).

Our findings on genomic surveillance in depicting local transmission and evolution of the D614G variant revealed that two of the variants had emerged locally: B.1.524, and AU.2, (S2 Table, highlighted in grey) here referred to as Malaysian lineages. Of these two, B.1.524, which was first detected in September, 2020, had silently caused the largest local transmission of the D614G variant in Malaysia (n = 419), followed by AU.2 (n = 311).

Furthermore, segregating the genomes analysis based on years, we found a clear pattern of lineage distribution which demonstrates how the major lineages disperse throughout Malaysia in 2020 and 2021 (S1 Fig). While the B.1.524 may have contributed heavily to the active spreading of D614G lineage locally, the data suggest that the AU.2 lineage, which was first detected on January 3, 2021 (S2 Table, highlighted grey), is currently taking its place as the major D614G variant contributor in spreading the disease. Using the patients' status metadata in S1 Table, we suggest that AU.2 might have originated from Sarawak, however, the origin of B.1.524 remains unknown.

**Table 1. The distribution of the lineages between gender, patient status and age groups.**

| | | Lineage, n(%) | | Total, n(%) | $C^2$ | p-value |
|---|---|---|---|---|---|---|
| | | AU.2 (GH) | B.1.524 (G) | | | |
| Gender | Male | 134 (37.9) | 220 (62.1) | 354 (100) | 3.862 | 0.049 |
| | Female | 132 (45.5) | 158 (54.5) | 290 (100) | | |
| Patient Status | Deceased | 0 | 3 (100) | 3 (100) | 1.541 | 0.463 |
| | Hospitalized | 0 | 6 (100) | 6 (100) | | |
| | Live | 36 (14.7) | 209 (85.3) | 245 (100) | | |
| Age Groups | 0–14 years | 27 (50.9) | 26 (49.1) | 53 (100) | 6.542 | 0.038 |
| | 15–64 years | 190 (40.7) | 277 (59.3) | 467 (100) | | |
| | >64 years | 43 (54.4) | 36 (45.6) | 79 (100) | | |

Next we analysed the association of gender, patient status and age group with Malaysian lineages B.1.524 and AU.2 (Table 1). Significant association was observed between lineages and both gender and age groups. Males have been found slightly more prone to be infected by the lineage, B.1.524 ($p$ = 0.049). It was also observed there is a significant association between lineages and age groups ($p$ = 0.038). However, there was no significant association observed between lineage and patient's status in term of disease severity.

## Origin of the massive spread of COVID-19 cases in Pahang

To infer the origin of the D614G variant that was responsible in causing widespread COVID-19 infections in Pahang this year, we built a NJ phylogenetic tree using complete genomes of D614G variant of Malaysian origin retrieved from GISAID along with sequence data from this study. The collection dates were restricted from January 1, 2021 to July 2021 (n = 986). Based on Nextstrain clade analysis, there were 22 SARS-CoV-2 lineages actively dispersed in Malaysia in 2021, divided into 10 clades (Fig 2A; S2 File). Genomes of Pahang SARS-CoV-2 D614G variants were found congregrated into African, Indonesian and Malaysian lineages (Fig 2B and 2C).

In order to further resolve the phylogenetic analyses of the D614G variant actively spreading in Pahang. A separate phylogenetic analysis was conducted with neigbouring representative sequences (in Nextstrain clade tree) of African, Indonesian and Malaysian lineages (Fig 2C) with Pahang SARS-CoV-2 D614G variants genomes. Topologically, the tree showed that EPI_ISL_2622079/SAMN19778017 (Kuala Lumpur, April 2021) clustered (100% bootstrap values) with B.1.462 (Indonesian lineage) with close relationship with EPI_ISL_2342564 (April 2021) and EPI_ISL_2090887 (May 2021), which were originated in Selangor (Fig 2C). Additionally, EPI_ISL_2622079 isolate was taken from subject having travel history from Kuala Lumpur to Pahang indicating interstate transmission of the Indonesian lineage of SARS-CoV-2 to Pahang (Fig 2B and 2C; S2 File). Whereas, EPI_ISL_2622089/ SAMN19778020 (Pahang, May 2021), EPI_ISL_2621677/SAMN19778012 (Pahang, April 2021) EPI_ISL_2622006/ SAMN19778019 (Pahang, April 2021), EPI_ISL_2622007/SAMN19778013 (Pahang, April 2021), EPI_ISL_2622046/ SAMN19778014 (Kelantan, April 2021), EPI_ISL_2622047/ SAMN19778015 (Kelantan, April 2021), EPI_ISL_2622088/SAMN19778018 (Kelantan, April 2021), clustered with β B.1.351 (African lineage) with 100% bootstrap values. It is important to note EPI_ISL_2622046 (Kelantan, April 2021), EPI_ISL_2622047 (Kelantan, April 2021), EPI_ISL_2622088 (Kelantan, April 2021) isolates were taken form subjects who had travelling history from Kelantan to Pahang and all three resided in different subclades of African lineage (Fig 2C) potentially indicating interstate transmisson of the virus. In comparison

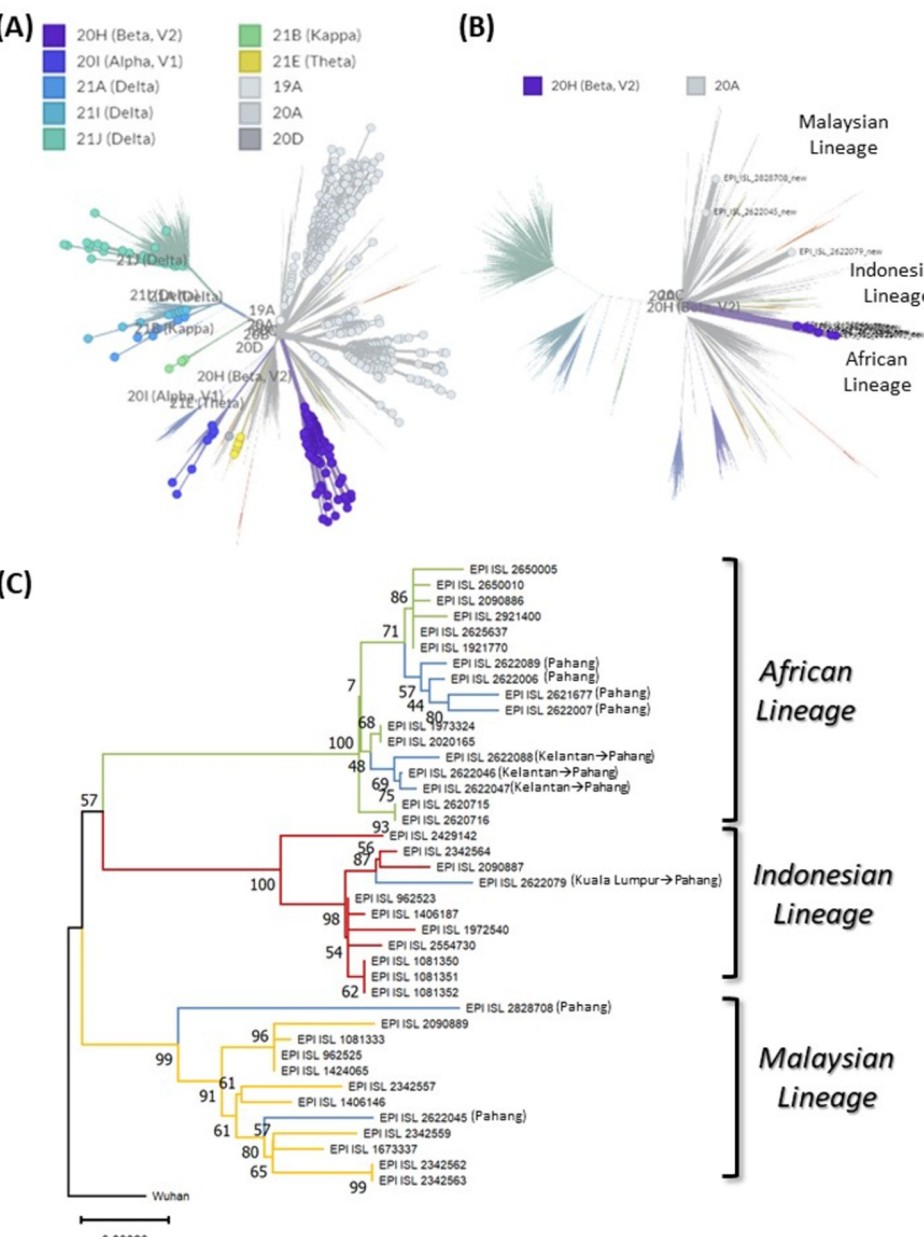

**Fig 2. Phylogenetic tree of 986 complete genomes of Malaysia SARS-CoV-2 D614G variants in 2021.** (A) Nextstrain clade distribution (inset) of 986 complete genomes of Malaysia SARS-CoV-2 D614G variant (filled circles) (B) Nextstrain clade and lineage distribution (inset) of the Pahang SARS-CoV-2 D614G variants (filled circle). Please see S2 File for the fully annotated tree (C) Phylogenetic relationship of the Pahang SARS-CoV-2 D614G variants (blue branches) with the selected neighbouring representatives (in Nextstrain clade tree) of SARS-CoV-2 genome sequences of African (green branches), Indonesian (red branches) and Malaysian lineages (yellow branches). The tree is reconstructed by Neighbor Joining (NJ) method with 1000 bootstrap replicates. Bootstrap values are indicated at nodes. Note the strong bootstrap support (99%-100%) at the common node of each lineage. In parenthesis city of the subjects are mentioned where arrow heads represent direction of traveling.

EPI_ISL_2828708/SAMN19778011 (Pahang, April 2021) and EPI_ISL_2622045/ SAMN19778016 (Pahang, April 2021), clustered (99% bootstrap values) with B.1.524 (Fig 2C), referred to as Malaysian lineage. All sequences are made avaiable at https://trace.ncbi.nlm.nih. gov/Traces/sra/?study=SRP324679.

## Mutations in spike protein of Malaysian lineages

Total 419 complete genomes of B.1.524 and 311 complete genomes of AU.2 were uploaded to Nextclade v.1.5.2 (https://clades.nextstrain.org) to analyse dominant mutations occur in spike protein of Malaysian lineages. In addition to D614G mutation, B.1.524 also carries A701V mutation in the spike protein. Whereas, AU.2 carries a mutation at positions N439K, P681R and G1251V.

## Amino acid mutations in spike protein of the Pahang- D614G variant SARS-CoV-2

Using Nextclade v.1.5.2 for clade assignment, mutation calling and sequence quality checks (https://clades.nextstrain.org), 1356 complete genomes of the D614G variant were analysed for sequence quality and mutations. Of that, 986 complete genomes passed Nextclade's sequence quality control highlighting different mutations in the spike protein. Our analysis revealed that all of Pahang's SARS-CoV-2 isolates has a unique substitution mutation of Glycine (G) to Cysteine (C) at position 1223 (G1223C) which was not found in the other 976 genomes (Fig 3).

The impact analysis of single-point mutations on protein-protein interaction binding affinity was performed using mCSM-PPI2. The result of the analysis are summarized in Table 2. To do this, a 3D structural model of wild type spike protein (YP_0097243901) was first generated through SWISS MODEL using a protein template model of 6XR8 (distinct conformation states of SARS-CoV-2 spike protein). Of note, mCSM-PPI2 is unable to predict the change in protein interaction affinity in deletions, hence analysis on L241del, L242del and A243del were not included in Table 2. To analyse the impact of G1223C mutation in the TM region of spike protein, a 3D structure model of the SARS-CoV-2 spike protein TM domain (7LC8) was retrieved from RCSB Protein Data Bank and was uploaded to mCSM-PPI2 server. Taken together, the missense mutations, L18F, N501Y, A701V and G1223C seem to have increased the binding affinity of the spike protein, whereas mutations D80A, D215G, K417N, N439K, E484K and A688S had the opposite effect. Unique mutation, G1223C, does not cause significant structural

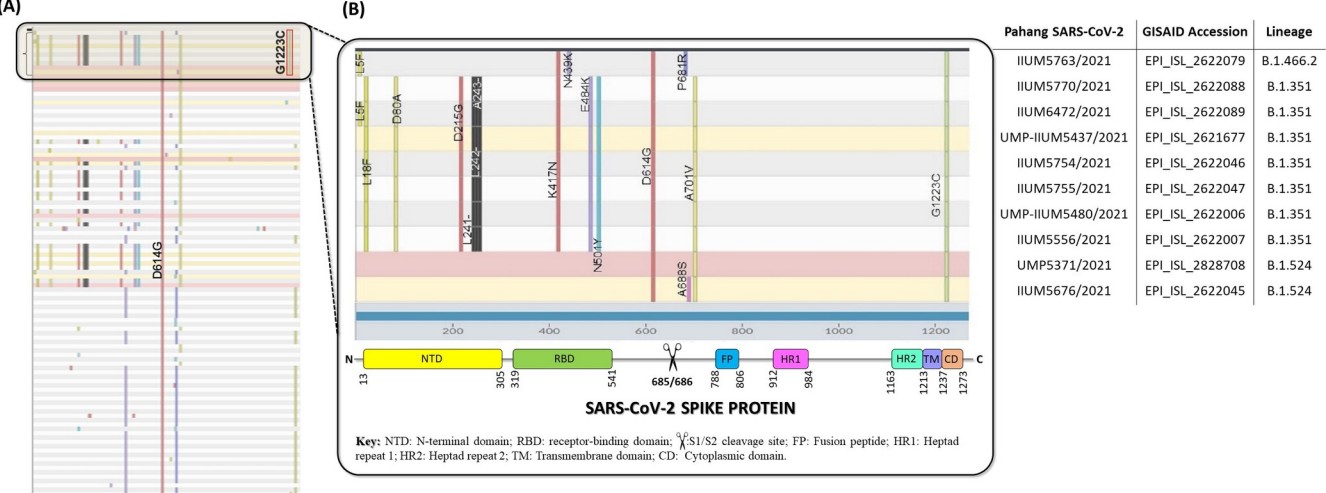

**Fig 3. Nonsynonymous mutations in the spike protein of Malaysian SARS-CoV-2 D614G variants.** (A) Nextstrain clade mutation analysis (vertical bars) of 986 Malaysian SARS-CoV-2 genomes, where Pahang SARS-CoV-2 D614G variants are highlighted in the box. Note the presence of unique mutation at G1223C in only Pahang SARS-CoV-2 D614G variants (B) Enlarged view of the Pahang SARS-CoV-2 D614G variants box where amino acid substitutions are annotated in the different regions of spike protein, schematically represented at bottom. Horizontal rows are correspondingly annotated with sample code, GISAID accession numbers and lineages.

**Table 2. The predicted effect of missense mutations in the spike protein of Pahang SARS-CoV-2 D614G variants.**

| Wildtype | Residue Number | Mutant | Distance to Interface | MCSM-PPI2-Prediction | Affinity |
|----------|---------------|--------|----------------------|---------------------|----------|
| Leu (L) | 18 | Phe (F) | 25.747 | 0.85 | Increase |
| ASP (D) | 80 | Ala (A) | 24.602 | -0.181 | Decrease |
| ASP (D) | 215 | Gly (G) | 23.94 | -0.178 | Decrease |
| LYS (K) | 417 | Asn (N) | 3.107 | -1.628 | Decrease |
| ASN (N) | 439 | Lys (K) | 7.043 | -0.319 | Decrease |
| GLU (E) | 484 | Lys (K) | 13.698 | -0.454 | Decrease |
| ASN (N) | 501 | Tyr (Y) | 7.588 | 0.16 | Increase |
| ALA (A) | 688 | Ser (S) | 26.327 | -0.044 | Decrease |
| ALA (A) | 701 | Tyr (Y) | 2.737 | 0.992 | Increase |
| GLY (G) | 1223 | Cys (C) | 5.489 | 0.233 | Increase |

rearrangement of the TM domain, except for the gain in salt bridge between C1223 and G1219 (Fig 4). In addition, PROVEAN and SNAP2 predicted decrease stability due to G1223C mutation in spike protein (Table 3).

## Discussion

The first incidence of COVID-19 in Malaysia was reported on January 25, 2020 and was traced back to three Chinese nationals who had direct contact with an infected individual while in Singapore [35]. The local Malaysian authority quickly developed standard guidelines for the management of COVID-19, including the set-up of designated hospitals and screening centers in each state [35]. To date, 2.6 million COVID-19 positive cases are recorded with over 30000 fatalities in the country. Based on earlier report, we found that SARS-CoV-2 variant with D614G mutation had been circulating in Pahang since April 2020 [25] and subsequently found elsewhere throughout Malaysia as the infection continues. For the record, the earliest study on SARS-CoV-2 virus genomes in Malaysia did not found D614G mutation, even the lineage B.6 that contributes profoundly in the second wave in Malaysia did not harbour D614G mutation in the spike protein [36].

Although major concerns have been raised on the emergence of VOC, SARS-CoV-2 β (B.1.351) and SARS-CoV-2 Δ (B.1.617.2), our analysis of SARS-CoV-2 genomes from the

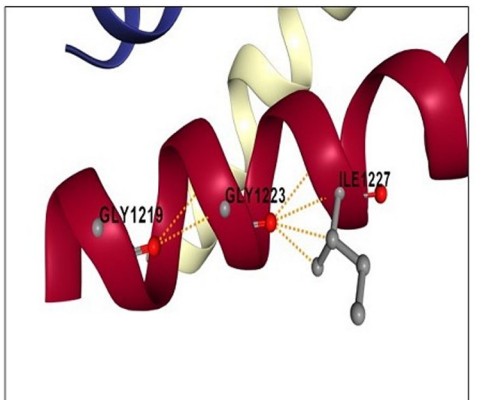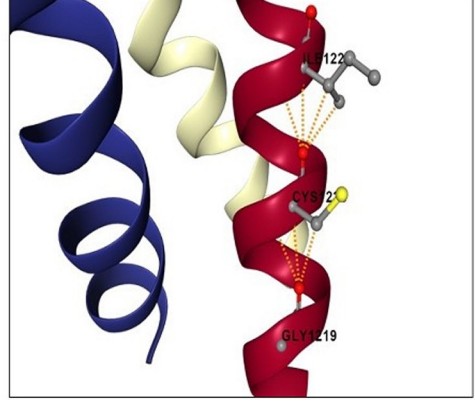

**Fig 4. Variations in the intramolecular interactions in transmembrane domain of wild type and mutant spike protein.** Intramolecular molecular interactions (yellow dotted line) in ribbon diagrams of the transmembrane domain of wildtype (G1223) and variant (C1223) SARS-CoV-2 spike protein.

**Table 3. Impact prediction of G1223C novel mutation.**

| Mutation in TM | mCSM-membrane (Benign/Pathogenic) | PROVEAN (Neutral/Deleterious) | SNAP2 (Neutral/Effect) |
|---|---|---|---|
| G1223C | Benign | Deleterious | Effect |

Malaysian population reported two different lineages of D614G variant that are actively dispersed locally. To our knowledge, the emergence of lineage B.1.524 was first detected in September last year. The analysis from early 2021 suggests that AU.2 had become the dominant lineage that actively spread in Malaysia in 2021 followed by B.1.524. We observed that AU.2 is closely related to B.1.4662 (Indonesian lineage), as both lineages carry the same amino acid mutations N439K in RBD and P681R in non RBD regions of spike protein. Its presence in Malaysia could be due to the spread of the disease *via* visitors from Indonesia [37]. Our analysis also suggests, AU.2 is not correlated to B.1.524, as B.1.524 carry different mutations (A701V) in spike protein. Moreover, lineages assignment using pangolin (v2.1.6, https://github.com/cov-lineages/pangolin) demonstrates prevalence of AU.2 and B.1.524 as Malaysia 94.0%, Indonesia 5.0%, United States of America 0.0%, India 0.0%, Singapore 0.0%, and Malaysia 76.0%, Singapore 16.0%, Thailand 3.0%, Philippines 2.0% and India 1.0%.

Of the 41 lineages of D614G variants detected in Malaysia since March 2020, 19 lineages have disappeared, leaving 22 lineages still actively spreading in 2021. During active propagation of the virus, new mutations accumulated in the progeny resulting in the emergence of new viral variants. Non-synonymous substitutions are extremely important since they result in an amino acid change, which may in turn induces structural change [38] and may then later have functional consequences in terms of transmission and pathogenicity [38]. In nations with poor containment capability, it was proven that the SARS-CoV-2 mutant lineage G (D614G) was able to replace earlier lineages more efficiently and was associated with a higher degree of disease severity [39]. Moreover, the emergence of more virulent strains such as included in VOC and VOI that harbored the D614G mutation in spike protein suggests that D614G variant had constantly subjected to positive selection pressure. Consequently, combination of various mutations in spike protein has been observed for increased viral transmission [40–42], increased disease severity [43], reduced susceptibility to the monoclonal antibody treatment [44] and reduced neutralization by convalescent and post vaccination sera [45–49].

Even though a recent study suggested that the GR was a predominant clade in Asia [50,51], our study found that GH is the major infecting clade in Malaysia, followed by G. To the best of our knowledge, studies related to AU.2 lineage in relation to disease epidemiology and pathology are scarce, however, the VOC, SARS-CoV-2-β (B.1.351) that grouped together with AU.2 clade, was reported to be linked with high disease severity and mortality [52]. Based on these reasons, we anticipated that this lineage may be the cause of high-risk transmission in Malaysia. On the other hand, the Malaysian lineage of B.1.524 is assigned to clade G that commonly associated with mild symptoms or asymptomatic cases [50]. Moreover, another study reported that the infection with clade G was not related with disease severity, and there was no clear indication of enhanced transmissibility despite greater viral loads [50].

Our metadata analysis showed higher ($p < 0.05$) prevalence among male patient with B.1.524 (G clade) variants, however, large-scale data is needed for further validation. Previous study has also shown that men with COVID-19 have relatively poor prognosis and mortality regardless of age [53] due to potential differences in the immune response between males and females [54]. The disease distribution is significantly higher among adolescent and adult age group in both AU.2 and B.1.524 group ($p < 0.05$). This could be explained due to presence of comorbidities, immunological senescence and changes in ACE2 receptor [50]. However, our

study showed no association between both lineages in relation to disease severity. Here, we anticipate that this may be due to lack of metadata related to the disease severity among Malaysian patients in GISAID database.

Higher infectivity of the SARS-CoV-2 variants is associated with increased in binding affinity between spike protein and ACE2 due to K417N, E484K, N439K and N501Y mutations in the RBD of the spike protein. While N501Y mutation alone enhanced spike RBD-ACE2 affinity [55], combination of K417N, E484K and N501Y mutations in B.1.351 lineage resulted in noticeable conformational changes in RBD when bound to ACE2 [56–58]. Although N439K mutation in RBD was first found in already extinct lineage B.1.1.41, a new lineage B.1.258 independently acquired the same amino acid substitution [59]. It is unknown whether B.1.466.2 (also known as Indonesian lineage) and AU.2 of Malaysian lineage, acquired N439K divergently and/or as a result of convergent evolution. Of concern, N439K mutation promotes evasion of antibody-mediated immunity by conferring resistance against several neutralizing monoclonal antibodies and reduces the activity of some polyclonal sera from patients recovered from infection [60]. However, there is no evidence of change in disease severity in a large cohort of patients infected with SARS-CoV-2 harbouring N439K mutation in the spike protein [60]. In addition, A701V mutation, adjacent to the furin cleavage site of spike protein subunit S1 and S2, in B.1.524 of Malaysian lineage was also found in SARS-CoV-2 β (B.1.351) strains and SARS-CoV-2 *i* B.1.526 (USA) [61]. Our computational analysis predicted A701V with increase protein-protein interaction affinity.

In tracking the distribution of the ten lineages which caused blooming of positive COVID-19 cases in Pahang this year, it appears that all virus collected from Pahang have the same substitution of amino acid at 1233 from Glycine to Cysteine in TM domain of spike protein, not found previously in Malaysia. While the significance of G1223C mutation is still unknown, it is well known that spike protein mediates entry of SARS-CoV-2 into target cells through two steps. First, it involves binding of RBD to its receptor, human ACE2, and is proteolytically activated by human proteases at the S1/S2 boundary. Second, S2 of spike protein including TM domain will undergoes structural change to mediate viral membrane fusion with the targeted cells [62,63]. To date, very little attention have been put on the TM domain involvement in the cellular entry of SARS-CoV-2. Although sequence analysis on TM domain among all coronaviruses spike protein conducted previously [61,63] revealed a high conservation rate in the region, however, extensive mutations in TM domain of SARS-CoV-2 caused incapability of the virus to establish complete membrane fusion process [63]. Highly conserved small amino acids in TM domain of SARS-CoV-2 spike protein (G1219, A1222, G1223, A1226) were initially thought to be important for TM domain oligomerization. However, recent findings showed neither glycine nor alanine in the trimer structure appeared to be important for hydrophobic core formation [64]. Thus, suggesting a possible role of the glycine motif is in a later step of fusion. We believe the effect of G1223C mutation in TM domain deserve further investigations in future functional experiments.

The present study has some limitations. First, the work on WGS in characterizing the circulating variants in Malaysia needed to be underscored systematically by representing Malaysian cases with considerably large sample size. Integration of viral genomics with the epidemiological and modelling data, local transmission chains and regional spread were able to be tracked and audited in real time [65]. This strategy was proven to curb the spread of COVID-19 in developed countries like Australia [66] and New Zealand [65]. Second, lack of metadata in GISAID database hampered the analysis of the impact of the distribution of individual clades on the localized disease epidemiology. We also discovered a plethora of unclear entries that offer very little information about the real source of the samples. All these issues can affect the effectiveness and accuracy of association studies. We therefore advocate for SARS-CoV-2

genomic data providers to provide comprehensive clinical details of deposited sequences, and also encourage genomic database maintainers to be aware of potential errors in incoming samples and to actively support metadata standards. One option may be to entirely disregard samples with suspected metadata issues, however, this may result in considerable reduction of sample size, thereby reducing the power of statistical studies [67].

## Conclusion

Herein, we have reported the most prevalent SARS-CoV-2 lineages of B.1.524 and AU.2 that sustained major outbreak of COVID-19 transmission during third wave of infection in Malaysia. Whereby the mutation at G1223C is under reported and further large-scale studies are warranted. Furthermore, the N439K mutation that observed in RBD of AU.2 deserves additional attention and monitoring due to its capability to increase virus infectivity while evading antibody-mediated immunity. Uncontrolled and intensive virus transmission will result in the emergence of new viral variants, which may significantly influence vaccine efficacy and perhaps, disease severity. The continuous emergence of novel SARS-CoV-2 variants highlights the need for public compliance with SOPs and other recommendations, notably mask use, hand cleanliness and physical separation, as well as the necessity to acquire herd immunity through the vaccination program. These measures will aid in slowing viral transmission and reducing the likelihood of new variations emerging in the SARS-CoV-2.

## Supporting information

**S1 Fig. D614G variant lineage distribution in 2020 and 2021 based on complete genomes deposited to GISAID (Malaysia).** A. The distribution of lineages from March to December, 2020. B. The distribution of lineages from January to July, 2020.
(TIFF)

**S1 File. Selected (987) sequences from SARS-CoV-2 D614G variant from Malaysia including Wuhan SARS-CoV-2.**
(DOCX)

**S2 File. Complete phylogenetic tree in MEGA XI format.**
(MTSX)

**S1 Table. Collate metadata.**
(CSV)

**S2 Table. Summary of date and D614G variant virus strains from each lineage first detected in Malaysia.**
(PDF)

## Acknowledgments

We acknowledge the COVID-19 task forces from Sultan Ahmad Shah Medical Centre @ IIUM and Universiti Malaysia Pahang, Malaysia.

## Author Contributions

**Conceptualization:** Mushtaq Hussain, Ayesha Aslam, Mohd Nazli Kamarulzaman, How Soon Hin, Hajar Fauzan Ahmad.

**Data curation:** Aini Syahida Mat Yassim, Mushtaq Hussain, Ayesha Aslam, Norhidayah Kamarudin.

**Formal analysis:** Ummu Afeera Zainulabid, Aini Syahida Mat Yassim, Mushtaq Hussain, Ayesha Aslam.

**Funding acquisition:** Ummu Afeera Zainulabid, Hajar Fauzan Ahmad.

**Investigation:** Ummu Afeera Zainulabid, Aini Syahida Mat Yassim, Mushtaq Hussain, Ayesha Aslam, Mohamad Shafiq Mohd Ibrahim, Hajar Fauzan Ahmad.

**Methodology:** Ummu Afeera Zainulabid, Mushtaq Hussain, Ayesha Aslam, Norhidayah Kamarudin, Hajar Fauzan Ahmad.

**Project administration:** Hajar Fauzan Ahmad.

**Resources:** Hajar Fauzan Ahmad.

**Software:** Aini Syahida Mat Yassim, Mushtaq Hussain, Ayesha Aslam.

**Supervision:** Mushtaq Hussain, Mohd Nazli Kamarulzaman, How Soon Hin, Hajar Fauzan Ahmad.

**Validation:** Ummu Afeera Zainulabid, Mushtaq Hussain, Ayesha Aslam, Mohamad Shafiq Mohd Ibrahim, Hajar Fauzan Ahmad.

**Visualization:** Mushtaq Hussain, Ayesha Aslam, Sharmeen Nellisa Soffian.

**Writing – original draft:** Ummu Afeera Zainulabid, Aini Syahida Mat Yassim, Sharmeen Nellisa Soffian, Hajar Fauzan Ahmad.

**Writing – review & editing:** Ummu Afeera Zainulabid, Aini Syahida Mat Yassim, Mushtaq Hussain, Ayesha Aslam, Sharmeen Nellisa Soffian, How Soon Hin, Hajar Fauzan Ahmad.

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
