## [Decision Letter · Decision Letter 0]

15 Oct 2021

PONE-D-21-25859Whole Genome Sequencing Analysis of Spike D614G Mutation Reveals Unique SARS-CoV-2 Lineages of B.1524 and AU.2 in MalaysiaPLOS ONE

Dear Dr. AHMAD,

Thank you for submitting your manuscript to PLOS ONE. After careful consideration, we feel that it has merit but does not fully meet PLOS ONE’s publication criteria as it currently stands. Therefore, we invite you to submit a revised version of the manuscript that addresses the points raised during the review process.

We look forward to receiving your revised manuscript.

Kind regards,

Ahmed S. Abdel-Moneim, Ph.D.

Academic Editor

PLOS ONE

Journal Requirements:

3. Please upload a new copy of Figure 2 as the detail is not clear. Please follow the link for more information: " ext-link-type="uri" xlink:type="simple">https://blogs.plos.org/plos/2019/06/looking-good-tips-for-creating-your-plos-figures-graphics/"
" ext-link-type="uri" xlink:type="simple">https://blogs.plos.org/plos/2019/06/looking-good-tips-for-creating-your-plos-figures-graphics/"

Reviewers' comments:

Reviewer's Responses to Questions

**Comments to the Author**

1. Is the manuscript technically sound, and do the data support the conclusions?

**Review Comments to the Author**

**Reviewer #1: General comment**

Considering the large scarcity of sequences from this country (but not limited to), the work of Zainulabid et al, about SARS-CoV-2 molecular epidemiology in Malaysia, comes to contribute with CoV-2 genomic surveillance studies. The work is logically well conducted and written. Considering the SARS-CoV-2 spread flow for other countries vs the intense individuals movement across the globe, the work also adds novelty by inferring and discussing the new lineages circulating. Although this Referee feels the lack of time-scale trees to try establish the origin of these new strains, on the other hand, considering the current pandemic scenario, the work brings interesting and useful data for the scientific community.   

Minor concerns

Materials and Methods

Sample selection

Please, inform us the protocol used for detection of SARS-CoV-2

RNAs extraction

Line 123: Please, add the ct value.

Public database SARS-CoV-2 genome analysis

Line 149: Does the criteria of sequence selection have been based only on the length? What about the sequencing coverage? Please, add such information.

Results

- Could you please clarify the results from Table 3? It seems paradoxical.

- Please, improve the quality of all figures presented.

- Fig. 2. Phylogenetic trees provided are difficult to read.

- Fig. 2 C and D - please, provide the whole tree.

- Fig. 3: What is A and B? Where is the start and end position? It is confusing. The way it is, it does not add too much to the work.

- Fig. 4. same above (Fig. 3)

- Fig. 5. Did you mean Spike TM Domain?

Origin of the massive spread of COVID-19 cases in Pahang

- Sometimes it is difficult to understand what the authors are inferring because of the codes such as IIUM5763/2021. Maybe replacing it by the city(district)/year (code between parenthesis) would become easier to the readers. The same for the trees (tip label).

- Please, replace and add the WHO new nomenclatures for SARS-CoV-2 variants since it becomes easier to understand --- in the whole manuscript.

Discussion

Line 350: Please, use “our analysis of” instead of “our detailed analysis of”.

Line 358: add a reference

Line 358: same from line 350.

Line 436: Please, choose to use countries or cities.

Reviewer #2: The paper is well written and results are novel. However, some grammatical errors needs the attention e.g. line 63 had (has), line 203 were (was), line 249 is (was), line 250 is (was), line 306 was (is), line 340 was (are), line 383 follows (followed) etc. Another concern is that the sample size is small. Please justify it. Add mechanism underlying the mode of action of SARS-CoV-2 in causing the disease and symptoms.

---

## [Author Response · Author response to Decision Letter 0]

14 Dec 2021

Journal Requirements:

Response: Thank you for providing the template. Our revised manuscript follows the journal guideline to the best of our understanding. 

Response: The data has been submitted to the GISAID public domain and accessible to all who wish to access. Accession numbers of sequences are also provided in the manuscript and in the phylogenetic analysis and figures. 

3. Please upload a new copy of Figure 2 as the detail is not clear. Please follow the link for more information: https://blogs.plos.org/plos/2019/06/looking-good-tips-for-creating-your-plos-figures-graphics/" https://blogs.plos.org/plos/2019/06/looking-good-tips-for-creating-your-plos-figures-graphics/"

Response: Figure 2 has been thoroughly revised for clarity and now represent more detailed and resolved information. 

Response: We have thoroughly scanned the list of references and no such reference (retracted) was found. We have also transformed the references as per PLoS One format. Citations of all papers which were then in preprint and now published are accordingly modified. 

Reviewers' comments:

Comments to the Author

Reviewer #1: General comment

Considering the large scarcity of sequences from this country (but not limited to), the work of Zainulabid et al, about SARS-CoV-2 molecular epidemiology in Malaysia, comes to contribute with CoV-2 genomic surveillance studies. The work is logically well conducted and written. Considering the SARS-CoV-2 spread flow for other countries vs the intense individuals movement across the globe, the work also adds novelty by inferring and discussing the new lineages circulating. Although this Referee feels the lack of time-scale trees to try establish the origin of these new strains, on the other hand, considering the current pandemic scenario, the work brings interesting and useful data for the scientific community. 

Response: We are grateful for the reviewer suggestion and in the phylogenetic analysis presented in the revised manuscript we have address this observation. 

Minor concerns

Materials and Methods

Sample selection

Comment: Please, inform us the protocol used for detection of SARS-CoV-2

Response: SARS-CoV-2 was detected through RT PCR and mentioned under Sample selection heading 

RNAs extraction

Comment: Line 123: Please, add the ct value.

Response: Samples with less than 25 ct values were selected for the subsequent investigation and this information has also been provided in the revised manuscript. 

Public database SARS-CoV-2 genome analysis

Comment: Line 149: Does the criteria of sequence selection have been based only on the length? What about the sequencing coverage? Please, add such information.

Response: Criteria of sequence selection has been elaborated and highlighted in the same section in the revised manuscript. “Sequences were selected based on completion of the genome with minimum number of unresolved nucleotides. Restraining the selection from the total of 1356 to 986 sequences”

Results

Comment: Could you please clarify the results from Table 3? It seems paradoxical.

Response: Table 3 shows the prediction of three different programs to suggest the potential impact of the mutations. Since all three programs based on different algorithms and used different base line data, it is possible, rather not very uncommon that they may end up giving different predictions for same mutation. 

Comment: Please, improve the quality of all figures presented.

 Fig. 2. Phylogenetic trees provided are difficult to read.

 Fig. 2 C and D - please, provide the whole tree.

Response: Figure 2 has been thoroughly revised for clarity and now represent more detailed and resolved information. Phylogenetic tree with more pertinent sequences was reconstructed with Wuhan as an outgroup with relatively more statistical support of 1000 bootstraps (revised version) compared to 500 (previous version). Whole tree is also provided in MEGA format for the reference as supporting information.

Comment: Fig. 3: What is A and B? Where is the start and end position? It is confusing. The way it is, it does not add too much to the work.

Response: We are highly grateful and agree to the suggestion of the reviewer, since the information is already present in the text of the manuscript, we have removed the figure 3. 

Comment: Fig. 4. same above (Fig. 3)

Response: We are highly indebted for the suggestion; Figure 4 has been modified for better clarity and respective legend of the figure elaborate the flow of the Figure 4 (previous version) and now Figure 3 (revised version). In the schematic diagram of spike protein N and C terminal are also labelled. 

Comment: Fig. 5. Did you mean Spike TM Domain?

Response: Yes, full name is also used in the figure legend 

Origin of the massive spread of COVID-19 cases in Pahang

Comment: Sometimes it is difficult to understand what the authors are inferring because of the codes such as IIUM5763/2021. Maybe replacing it by the city(district)/year (code between parenthesis) would become easier to the readers. The same for the trees (tip label).

Response: Indeed, this is an excellent suggestion and we have made two important changes in this regard.

1. In the revised manuscript and in the phylogenetic analysis figure (Fig 2) we have used GISAID accession number instead of sample code along with the geographical location of Pahang SARS-CoV-2 D614G variants. 

2. In the revised manuscript we have also added the collection month and year of the Pahang SARS-CoV-2 D614G variants

Comment: Please, replace and add the WHO new nomenclatures for SARS-CoV-2 variants since it becomes easier to understand --- in the whole manuscript.

Response: In the revised manuscript WHO nomenclature is used where appropriate 

Discussion

Comment: Line 350: Please, use “our analysis of” instead of “our detailed analysis of”.

Response: Modifications made 

Comment: Line 358: add a reference

Response: Reference added 

Comment: Line 358: same from line 350.

Response: Modifications made 

Comment: Line 436: Please, choose to use countries or cities.

Response: We are grateful to the reviewer for the suggestion and modifications are made in the revised manuscript 

Reviewer #2: The paper is well written and results are novel. However, some grammatical errors needs the attention e.g. line 63 had (has), line 203 were (was), line 249 is (was), line 250 is (was), line 306 was (is), line 340 was (are), line 383 follows (followed) etc. 

Response: We are thankful to the reviewer for the suggestion and modifications are made in the revised manuscript 

Comment: Another concern is that the sample size is small. Please justify it.

Response: Sample selection criteria has been explained in the methodology section of the revised manuscript. 

Comment: Add mechanism underlying the mode of action of SARS-CoV-2 in causing the disease and symptoms.

Response: We are thankful to the reviewer for highlighting this, brief description regarding the SARS-CoV-2 mode of action has been added in the introduction.

---

## [Decision Letter · Decision Letter 1]

25 Jan 2022

Whole Genome Sequence Analysis Showing Unique SARS-CoV-2 Lineages of B.1.524 and AU.2 in Malaysia

PONE-D-21-25859R1

Dear Dr. AHMAD,

We’re pleased to inform you that your manuscript has been judged scientifically suitable for publication and will be formally accepted for publication once it meets all outstanding technical requirements.

Kind regards,

Ahmed S. Abdel-Moneim, Ph.D.

Academic Editor

PLOS ONE

---

## [Editor Report · Acceptance letter]

18 Feb 2022

PONE-D-21-25859R1 

Whole Genome Sequence Analysis Showing Unique SARS-CoV-2 Lineages of B.1.524 and AU.2 in Malaysia 

Dear Dr. Ahmad:

I'm pleased to inform you that your manuscript has been deemed suitable for publication in PLOS ONE. Congratulations! Your manuscript is now with our production department. 

Kind regards, 

on behalf of

Prof. Ahmed S. Abdel-Moneim 

Academic Editor

PLOS ONE